# Climate Change, Drought and Rural Suicide in New South Wales, Australia: Future Impact Scenario Projections to 2099

**DOI:** 10.3390/ijerph19137855

**Published:** 2022-06-27

**Authors:** Ivan C. Hanigan, Timothy B. Chaston

**Affiliations:** 1WHO Collaborating Centre for Environmental Health Impact Assessment, School of Population Health, Faculty of Health Sciences, Curtin University, Perth, WA 6102, Australia; 2University Centre for Rural Health, Sydney School of Public Health, The University of Sydney, Sydney, NSW 2006, Australia; timothy.chaston@sydney.edu.au; 3Environment Protection Authority Victoria, Melbourne, VIC 3001, Australia

**Keywords:** wellbeing, mental health, rainfall, drought, climate change scenario, vulnerable populations

## Abstract

Mental health problems are associated with droughts, and suicide is one of the most tragic outcomes. We estimated the numbers of suicides attributable to drought under possible climate change scenarios for the future years until 2099, based on the historical baseline period 1970–2007. Drought and rural suicide data from the Australian state of New South Wales (NSW) were analyzed for the baseline data period. Three global climate models and two representative concentration pathways were used to assess the range of potential future outcomes. Drought-related suicides increased among rural men aged 10–29 and 30–49 yrs in all modelled climate change scenarios. Rural males aged over 50 yrs and young rural females (10–29) showed no increased suicide risk, whereas decreased suicide rates were predicted for rural women of 30–49 and 50-plus years of age, suggesting resilience (according to the baseline historical relationship in those population sub-groups). No association between suicide and drought was identified in urban populations in the baseline data. Australian droughts are expected to increase in duration and intensity as climate change progresses. Hence, estimates of impacts, such as increased rural suicide rates, can inform mitigation and adaptation strategies that will help prepare communities for the effects of climate change.

## 1. Introduction

Droughts have undermined community resilience and have contributed to declines and even collapse of some civilizations in the past [1]. Anthropogenic greenhouse gas emissions have already led to increased drought frequencies during the twentieth century [2], and future droughts are likely to be more frequent and severe [3,4,5], especially in Australia [6,7].

Droughts have been associated with distress and depression [8,9,10,11] and with suicide rates [12,13]. Before the Covid pandemic struck, Australia was already suffering high suicide rates. In 2017, the pre-pandemic suicide rate was 10.4 per 100,000 globally but was 12.8 per 100,000 in Australia (Source: Institute for Health Metrics and Evaluation, available from http://ghdx.healthdata.org/gbd-results-tool, accessed on 14 June 2022), warranting concern about the additional impact of drought on mental health in Australia. Because suicide affects whole communities, the relationship between drought and suicide has received substantial public interest in Australia, and it is current standard practice that political and media statements relating to drought and suicide are accompanied by phone numbers for mental health support organizations, such as the Australian support service Lifeline (www.lifeline.org.au, accessed on 14 June 2022).

In a systematic review of studies reporting relationships between drought and health [14], several potential pathways were identified, including specific economic impacts and effects on migration and social cohesion (also supported by other investigations [15,16]). However, the potential for drought-related suicide rates to increase under climate change scenarios has not been assessed.

There are several theories about how climatic drought may influence the suicide rate, as explored in previous review articles [15,17]. First, droughts increase financial stress on farmers and farming communities, even if partially compensated by drought relief welfare payments. Farmers are also more vulnerable to economic realities, such as rising interest rates, falling commodity prices and unfavorable foreign exchange rates. Reduced rainfall can directly depress economic activity in rural towns. In some regions, the entire economy may be affected. Rural downturns can accelerate migration to metropolitan areas; weakening and stressing social support systems and lessening social interaction. In some cases, rural depopulation may pass a tipping point, leading to an ongoing loss of critical services, such as hospitals, schools and doctors. Second, there can be a great psychological toll following environmental degradation and this may be acute during droughts, especially when decisions are made to sell or kill starving animals or to destroy orchards and vineyards, which in some cases were painstakingly accumulated over generations. Such loss, and even the apprehension of loss, undoubtedly places a burden on the mental health of farmers and their families. This mourning may not be confined to farmers because other sections of the community are also likely to be impoverished by long-term environmental degradation. The experience of seeing suffering wild plants and animals, or parched urban parks and gardens, and contemplation of their loss are likely to be extremely painful for some individuals.

In this study, we estimated the number of drought-attributable rural suicides in NSW under six projected climate change scenarios out to 2099. To this end, we calculated distributions of future drought durations under the conditions of three accepted climate change models at two representative concentration pathways (RCP) and then applied an exposure-response function for drought and suicide that was calculated from 38 years (1970–2007) of age- and sex-stratified climate and suicide records in the Australian state of New south Wales (NSW). The resulting projections indicate that suicide rates in rural men of working ages will become a major public health concern.

## 2. Materials and Methods

We followed the climate change impact assessment method described in a methodology paper for climate change health impact assessments by Vicedo-Cabrera et al. [18]. Briefly, historical records were used to quantify the association of climate with health, and the estimated relationships were projected to chosen future times based on climate change models. To calculate drought durations during the future years to 2099 in NSW, we used three general circulation models (GCM) of climate change-related changes in monthly rainfall and two RCPs.

The drought–suicide association was applied to drought projections from climate variables only, assuming no migration or adaptation or demographic changes, as described in the methodological framework [18]. This approach therefore indicates the impacts of drought under the assumption of all else being equal, and that none of the other complex factors influencing suicide rates are included (to avoid over-complicated scenario modelling but instead follow a “climate analog” approach).

### 2.1. Study Region 

The southeast Australian state of NSW has often experienced severe droughts, and these are expected to increase [7]. We divided NSW into 11 statistical divisions (SDs) from the Australian census (Figure 1). These study areas are distinguished on the basis of climate, topography and agricultural production. The SDs Sydney, Illawarra and Hunter are classified as urban based on the sizes of the respective major cities Sydney, Wollongong and Newcastle. Among the 8 rural SDs, North West and Far Western have small populations and were aggregated to yield more stable suicide rates for statistical modelling.

NSW had a population of around 8 million in 2018, with approximately 5 million urban dwellers in the large coastal cities Sydney, Newcastle and Wollongong and the remainder in rural farming communities. The average rainfall in rural SDs of NSW during the years 1971 and 1999 was 771 mm (range across SDs: 371 to 1405 mm per year). Rainfall varies across the state, with dryer conditions in the far northwest and wetter conditions in the east. Rainfall has been exceptionally low in some NSW SDs over recent decades, and the widespread drought events during the period 2001–2009 are collectively known as the ‘Millennium Drought’ in recognition of their extremity [7].

### 2.2. Hutchinson Drought Severity Index

The Hutchinson Drought Severity Index (HDSI) is an Australian indicator of agricultural droughts that is based solely on rainfall data [19]. This index successfully predicted 69% of NSW government-declared droughts. The HDSI is a measure of extreme and protracted periods of dryness, benchmarked against historical rainfall data from no less than 30 years of records, and preferably from 50 to 100 years. HDSI scores are derived from the ‘Hutchinson Score’ [19]. The Palmer Index, which informed Hutchinson’s design, used a soil moisture balance equation to assign scores of −4 to +4 to represent drier and wetter conditions, respectively. In contrast, the Hutchinson Score expresses rolling rainfall totals for the six months up to and including the nominated month as percentiles relative to rainfall totals for corresponding sequences of six months over the historical record. Hence, the 6-month rainfall total of a nominated January is ranked to show dryness relative to all other January’s in the record. After linear rescaling, the 1st, 50th and 99th percentiles correspond with indexes of −4, 0 and 4, respectively. A score of 0 indicates that rainfall during the selected six-month period exceeded the long-term average for the same six-month period no more than 50% of the time. A Hutchinson Score of −1 corresponds with Palmer-Index definition of the start of mild drought, which becomes a full drought if it meets a second threshold. Breach of this threshold invokes the declaration of full drought. 

The HDSI accounting procedure creates a composite measure from which drought durations can be calculated. Whereas six consecutive months of mild drought is within the range of normal variability, droughts of seven or more months have been correlated with government drought declarations data [19]. Accordingly, month seven of relatively dry mild drought is considered the first month of full drought, and the end of the drought period is declared when the Hutchinson Score for a month rises above −1. Counting toward the seven-month dryness threshold for a new drought period re-starts when the Hutchinson score again falls to or below −1.

In the original formulation of the HDSI, scores were based on deviations of precipitation from historical calendar-month-specific averages for each region. To enable comparisons across changing climate regimes over the entire 130-year study period (1970–2099), we derived percentiles using a rolling 30-year look-back period in which drought indices were computed for the nominated year with the previous 29 years. As such, the computations represent the range of drought conditions in living memory of adults aged 30–49, so that perceptions of normal climate conditions vary with changes in rainfall and drought patterns. Additional details of the changes we made to the Hutchinson Drought Severity Index method are presented in the Appendix A. 

### 2.3. Rainfall and Temperature Data Inputs

To calculate drought indexes, we aggregated gridded monthly rainfall and temperature data from 1940–2007 to SDs using spatially weighted averaging over the polygons. Weather data in 5 × 5-km grids were obtained from the Australian Water Availability Project (AWAP Australian Bureau of Meteorology, Bureau of Rural Sciences, and CSIRO, http://www.bom.gov.au/jsp/awap/, accessed on July 2018). Full details of the AWAP data are provided in a previous report [20]. Gridded climate data were downloaded and prepared using open-source software tools in R (https://github.com/swish-climate-impact-assessment/awaptools, accessed on 14 June 2022).

### 2.4. Population Data

Population census data was used to adjust suicide rates for different sized denominator populations in SDs (Figure 1). Population estimates were derived for intercensal years by linear interpolation between census counts for 1971, 1976, 1981, 1986, 1991, 1996, 2001 and 2006 from the Australian Data Archives. Geographical boundaries of Australian Bureau of Statistics SDs 2006 were downloaded on 9 January 2012 from http://www.abs.gov.au/AUSSTATS/abs@.nsf/DetailsPage/1259.0.30.0022006?OpenDocument.

### 2.5. Suicide Data

All Australian deaths from self-inflicted injuries are assessed by a coroner to determine if they are suicide [21]. We extracted age- and sex-specific deaths for NSW from the Australian Bureau of Statistics (Causes of Death Australia (3303.0); data available on request). Causes of death were coded using the International Classification of Diseases (ICD) system, which was revised three times during the study period. The codes for suicide and intentional self-inflicted injury were E950.0–E959.9 in ICD-8 (used in 1970–1978) and ICD-9 (used in 1979–1996) and X60–X84.9 and Y87.0 in ICD-10 (used from 1997 to 2007). Data were aggregated at the scale of age groups, sex and by geographical regions bounded by ABS SD 2006. Further details are presented under “Suicide Data” in the Appendix A. Ethical approval was granted for this study.

### 2.6. Climate Change Scenario Projections

Data from three GCMs were chosen as a representative set of models from Phase 5 of the Coupled Model Inter-comparison Project (CMIP5) available from the CSIRO and Bureau of Meteorology, “Climate Change in Australia” dataset (https://doi.org/10.4225/08/55945F739A66D, accessed on 14 June 2022). These modelled data estimate rainfall by month between 2006 and 2100 as the percentage change compared to the 1986–2005 baseline climate. Assessments by the CSIRO and Bureau of Meteorology [22] found that the ACCESS1.0 model had maximum consensus with the other models for many regions, whereas the GFDL-ESM2M model was hotter and drier for many regions, and the NorESM1-M model was representative of a low warming, wetter scenario. These models represent the current south-east Australian climate and were used as suggested by Harris et al. [23]. RCP4.5 and RCP8.5 represent higher- and lower-emission scenarios, respectively. The baseline for climate change projections was 1986–2005.

Projection data were aggregated to ABS SD 2006 census boundaries. We generated average monthly rainfall projections for SDs by taking spatially weighted averages of pixels from each GCM that intersected each spatial boundary. More details of the data used for this study are presented in the corresponding Appendix A.

### 2.7. Statistical Modelling

Baseline exposure–response associations were quantified using time-series models developed using variables that were selected from a previously published best fitting model [13], in which potential interactions between drought and temperature, including non-linear effects, were investigated. Drought durations were log transformed. To account for small numbers and many zeros, we tested monthly Poisson and quasi-Poisson time series generalized linear models (GLMs) and generalized additive models (GAMs) and explored the potential interactions with temperature anomaly by region, age and sex. The dispersion parameter for the quasi-Poisson model was close to one, indicating no overdispersion. We used the generalized cross-validation (GCV) tool in the MGCV package of R to automatically estimate the appropriate shape of subgroup response functions.

The best fitting model was Poisson and had the following terms:
$$\begin{array}{cc} log\left(O_{ijkt}\right)= & s\left(Drought_{kt}\times Sex_{j}\times AgeGroup_{i}\times RuralOrUrbanRegion_{r}\right)\\ & +AgeGroup_{i}\times Sex_{j}\times s\left(Time_{t},df=3,basis=NaturalCubicSpline\right)\\ & +StatisticalDivision_{k}\\ & +tmax\_anomaly_{kt}\\ & +s\left(Month_{m},df=4,basis=CyclicCubicSpline\right)\\ & +offset\left(log\left(Pop_{ijkt}\right)\right), \end{array}$$
where $O_{ijkt}$ = monthly suicide counts by $AgeGroup_{i}$ (for the age group categories 10–29, 30–49 and 50 plus), $Sex_{j}$, $StatisticalDivision_{k}$ and $s\left(Drought_{kt}\times Sex_{j\;}\times AgeGroup_{i\;}\times RuralOrUrbanRegion_{r}\right)$ are interaction effects that may be linear or non-linear. $Time_{t}$ is a continuous variable for each month in sequence from January 1970 until October 2007. Month is the calendar months of the year, ranked from 1 to 12 and *s*($Month_{m},df=4,basis=CyclicCubicSpline$) represents a cyclic function for season, where ‘df’ denotes the degrees of freedom allowed for this term. 

The *s*(…) term represents a penalized regression spline for which the df may be specified or estimated from cross-validation using GCV. The term $tmax\_anomaly_{kt}$ is the monthly anomaly for the monthly average of daily temperature maxima compared to the long-term average. $Pop_{ijkt}$ is the interpolated population by month in each group and was calculated using linear interpolation between the census counts provided at 5-year intervals. The final model had all linear terms for rural subgroup associations between suicide and drought. 

Attributable number (AN) calculations of future drought associated suicide deaths were only performed for months in full drought (drought index > 6) and were expressed per annum by age and sex (only for rural SDs) using the following equation:$${\mathrm{AN}}_{ijk}=\overset{kt}{\sum }(1-e^{\left(-\beta_{ij}\times X_{kt}\right)})\times HistoricalNumber_{ijkm}$$
where *_k_* denotes each rural SD (or *zone_k_*), *t* is time (specified at each future *month_t_*), $\beta_{ij}$ is the exposure variable coefficient for $age_{i}$ and $sex_{j}$, $X_{kt}$ represents projected future drought exposure variables in $zone_{k}$ at *Time_t_* (in months) and $HistoricalNumber_{ijkm}$ is average deaths in $month_{m}$ and $zone_{k}$ of $age_{i}$ and $sex_{j}$.

We estimated empirical confidence intervals for 95% coverage from a Monte Carlo analysis based on 1000 sampling iterations from a distribution around the β coefficient and estimated standard error from the linear Poisson model described above.

R version 3.4.4 was used for all data preparation and analysis (R Core Team 2018. R Foundation for Statistical Computing, Vienna, Austria. URL https://www.R-project.org/).

## 3. Results

Summary descriptive statistics of annual rainfall, annual average monthly maximum temperatures, drought frequency and drought duration across SDs are shown in Table 1 for the historical period 1971–1999.

Descriptive statistics of suicide rates and populations in ABS SDs over the historical period 1971–1999 are shown in Table 2. 

In the historical period 1971–1999, the monthly maximum temperature anomalies were not correlated with the drought index (Pearson correlation 0.31; Spearman correlation 0.33). As shown in Figure 2, historical drought durations were greatest in centre and north-east SDs and under the higher emissions scenarios of RCP8.5. Yet, decreased drought durations are projected for some areas of the state. In the lower emissions scenarios of RCP4.5, increased durations are expected in most areas.

Time-series modelling of historical data showed a linear relationship between drought and suicide among rural males aged 30–49 yrs, with a relative risk (RR) of 1.14 (95% confidence intervals (CI), 1.07–1.22; *p* < 0.0001) for interquartile-range (IQR) increases in the drought index (1.07 on the log scale, or about 2 months). Among rural females of 30–49 years, however, the risk of suicide decreased with IQR increases in drought durations (RR 0.86 per IQR, 95% CI, 0.80 to 0.91; *p* < 0.05), and the causes of this inverse association remain unknown. A drought-associated risk was also identified among rural males aged 10–29 yrs (*p* < 0.01). All of these associations were absent in corresponding age and gender groups of urban populations (data not shown).

In additional sensitivity analyses, we allocated increased df to *Time_t_* and *Month_m_* spline terms, and also tested the use of sine and cosine terms to capture seasonal cycles in suicide rates. The results from sensitivity analysis are listed in the Appendix A “Sensitivity testing for splines of time”. These additional analyses did not change the associations in the main models, confirming that drought has no effect on urban suicide rates and that the effects of drought on rural suicide rates are linear. Thus, all further analyses were performed using a linear Poisson model. 

The present historical and climate change models indicate significant numbers of drought-attributable suicides in rural men aged 10–29 and 30–49, and in climate change scenarios with increased average drought durations, suicide rates will likely be exacerbated (Figure 3). Drought-attributable suicide rates among older rural males (>50 yrs) and young rural females (10–29 or 30–49 yrs) were not different in historical or climate change models, but suicides among older women (50 plus) may decrease in droughts.

The point estimates represented by the black dots in Figure 3 show the projected AN of suicide deaths per annum during the two periods 1971–1999 (historical) and 2000–2099 (climate change scenarios). The estimated attributable number of excess suicides in full drought for rural males aged 30–49 was 0.82 (95% CI 0.45, 1.15) per annum in the historical period and was 1.51 (95% CI 0.84, 2.13) per annum under the driest scenario GFDL-ESM2M RCP4.5 (a percentage increase of 84%, 95% CI 2, 159). Concomitantly, annual drought attributable suicides among women of this age group were −0.29 (95% CI −0.61, −0.03) and −0.53 (95% CI −1.14, −0.05), respectively. In the historical period, drought attributable suicides differed significantly (*p* < 0.05) by 1.11 (95% CI 1.06, 1.18) per annum between middle-aged men and women. In the driest scenario, GFDL-ESM2M RCP4.5, however, projected drought attributable suicide rates differed significantly by 2.04 (95% CI 1.98, 2.18; *p* < 0.05) between men and women of this age group. No gender differences in drought-related suicide rates were found in urban populations. The predicted number of annual suicides in middle-aged males attributable to periods of full drought accounted for around 2% of the total number of suicides in that group in rural regions, compared with 0% in urban regions.

## 4. Discussion

The results of our drought and suicide models suggest that future changes in drought durations, estimated across six scenarios (three GCMs and two RCPs), will increase suicide rates among males in rural NSW 2000–2099 compared with the historical experience of 1971–1999, although in many scenarios this was not statistically significant. Counter-intuitively, the excess risk of suicide in rural females was projected to decrease with increasing drought durations. 

A significant relationship between drought severity and suicide rates among working-age rural males in NSW over the period 1971–2007 was found previously [13]. However, a tendency for drought-related decreases in suicide rates among women of the same age group was observed. To further investigate the public health relevance of these observations, we calculated future drought durations in SDs of NSW using projected rainfall predictions from three GCMs with two RCPs (Figure 2). We then applied the exposure-response function for drought severity and suicide to predict drought-related suicide rates in rural NSW for the period 2000–2099. As shown in Figure 2, average drought durations may be expected to increase with climate change across the SDs of NSW. 

In accordance with our previous findings from drought and suicide records, the present climate change projections of increased average drought severity will be accompanied by divergence of suicide rates in men and women (Figure 3). Due to the wide distributions of future drought severity, historical ANs did not differ statistically significantly from any of the projected ANs, despite an 84% increase for annual deaths of rural males under the driest climate change scenario (GFL-ESM2M). Yet drought-ANs of suicides were significant in all models. Moreover, historical ANs differed significantly between men and women of working ages, and differed significantly more under the GFL-ESM2M scenario (Figure 3). 

Droughts combine the effects of reduced rainfall, high temperatures, low humidity and high evaporation and can lead to increased severity of bushfires (wildfires), dust storms and heatwaves. The mental health impacts of increasing temperatures have been studied previously [24,25], and key pathways through which drought could affect mental health and suicide rates have been identified in multiple studies [15]. In a systematic review of the literature, Vins et al. [15] found that financial stresses on farming families can substantially undermine mental health, even when partly compensated by drought relief welfare payments. That review also found that droughts accelerate migration from affected areas, thus weakening social support networks and decreasing social interactions, potentially leading to mental health impacts from the ongoing loss of regional hospitals, schools and industries [15]. 

For comparison of the magnitude of the estimated changes, we can contrast our study with those of Guo et al. (2018) [26], who predicted a 100% increase in heatwave-related excess deaths in Australia for the future period 2031–2080, or similarly, Physick et al. (2014) [27], who predicted that 2.3–27.3% increases in mortality will follow climate change-related increases in atmospheric ozone in Sydney out to 2060. The present study is the first to project drought-related mental health outcomes in Australia. Several studies from other countries report impacts of climate change-related air pollution and temperature on cardiovascular and respiratory mortality and morbidity, as reviewed by Lou et al. (2019) [28]. These studies generally project increased impacts under global warming scenarios. 

Our conclusions are highly transferable to environments where drought severity will increase and the livelihoods of rural communities are directly dependent on agriculture. In particular, where farming males’ identities are strongly linked to the productivity of their farms, suicides will be strongly linked to drought. Some aspects of this study may, however, be peculiar to the Australian context, such as reduced numbers of female suicides during drought. Perhaps this observation is specific to rural Australian women and the cultural and social contexts of specific communities. Hence, because future studies of specific rural communities may reveal different gender-based relationships, extrapolation of this finding to other countries should be performed cautiously and should be supported by baseline data from location-specific studies. 

There are several possible explanations for the contrasting effects of drought on suicide rates among working males and females of the same Australian population. First, rural women seek diverse social supports that relieve stress whereas men may be too ashamed [29]. In an application of the suicide theories of Durkheim to farming communities, Hogan et al. suggested a male reluctance to adjust anomic and egoistic misconceptions in the face of identity shock [30]. Australian rural women may also be more stoic and resilient in the face of drought-related hardships. Furthermore, community support may strengthen as drought conditions deteriorate, reinforcing the support networks that benefit rural women most. In addition, the Australian government has previously offered financial support following drought declarations which may be beneficial for rural women but not for their male counterparts due to machismo [30]. Suicide rates are higher in farming communities than in urban areas in Australia [30,31,32] and in other countries [33]. In a review of this phenomena [31], it was argued that social, geographical and psychological factors together contribute to this disparity between rural and urban communities.

Strengths of this study include the use of a long period of comparable and exhaustive mortality and drought data and the validated HDSI drought index. A further strength is that the GCM projections were all identified as high-quality by the CSIRO using the Climate Futures tool [22].

Limitations of this study include a potential exposure mis-classification bias due to the scale of the GCM and the limited assessment of regional patterns in the modelled climate data. Another limitation of this study is that we did not consider the well-known effect of temperature on suicide [24], although no interaction with the effects of drought on suicide were identified and therefore may not undermine the assessments herein. Nevertheless, a key consideration for future studies should be to include temperature and to explore interactions with humidity because drought indicators are sensitive to evaporation and humidity as well as rainfall.

## 5. Conclusions

This study alludes to a suite of contributing factors that influence suicide, drawn from the environmental, social and political context of life in Australia, of which drought is a part. These results help isolate the most critical times of risk and could be used to better time and direct the deployment of social resources. This includes provision of targeted counselling services to vulnerable people, both during droughts and at times with hotter than average maximum temperatures. Other policy implications from this finding support broadening investment in research into gender-specific drought effects rather than purely climate and economic focused research into drought impacts.

In conclusion, large and rapid changes in rainfall patterns in south-east Australia are expected over the next 80 years, with likely increases in drought duration and severity. The present analyses predict increased suicide rates in rural men compared with women. Alternately, suicide rates in older males and younger females were not predicted to change, although a decrease was estimated for rural older women. With further estimates of mental health outcomes, this assessment of rural suicides will help prepare Australian society for the effects of climate change and may inform global efforts to mitigate and adapt to environmental changes.

## Figures and Tables

**Figure 1 ijerph-19-07855-f001:**
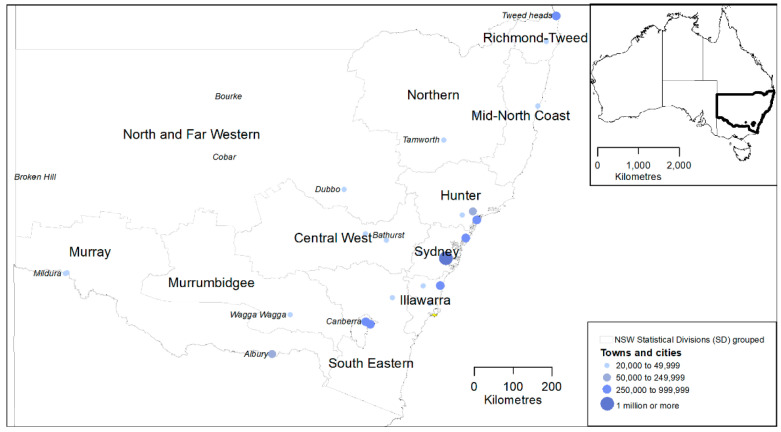
Statistical divisions (SDs) of NSW from the 2006 edition of the Australian census geography; the small populations of North West and Far Western SDs were combined to support statistical models. Population centres and selected rural farming communities are also shown for context.

**Figure 2 ijerph-19-07855-f002:**
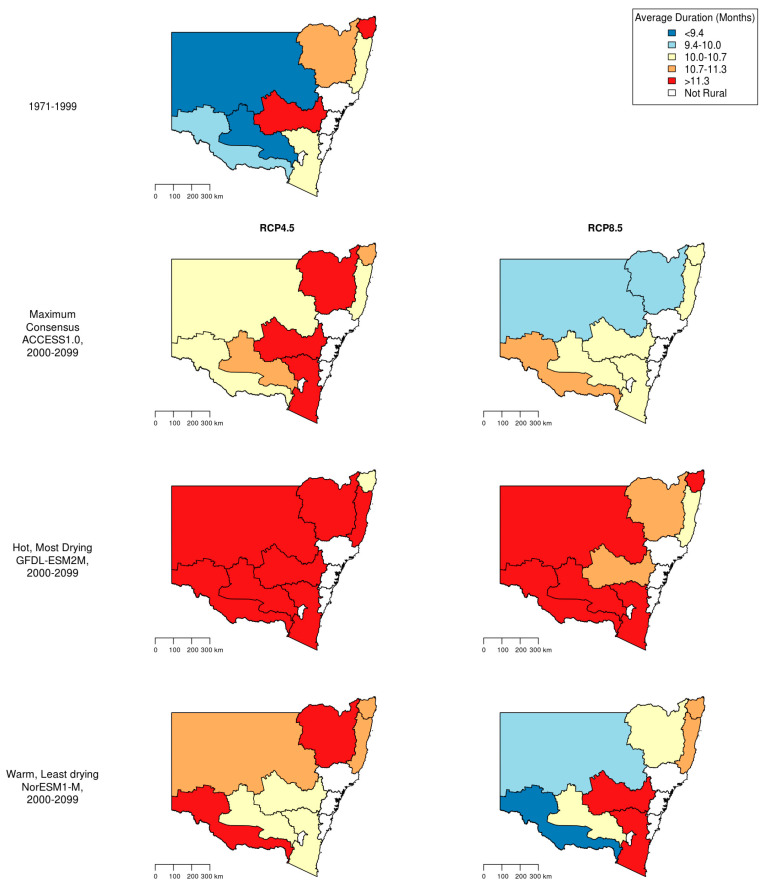
Average drought durations in rural NSW statistical divisions (SDs) during 1971–1999 and in future general circulation models (GCM) and representative concentration pathway (RCP) scenarios (2000–2099); RCP8.5 and RCP4.5 represent high and lower emissions, respectively. ACCESS1.0 was the model with maximum consensus, GFDL-ESM2M was the hotter and drier model and NorESM1-M was the low warming, wetter model.

**Figure 3 ijerph-19-07855-f003:**
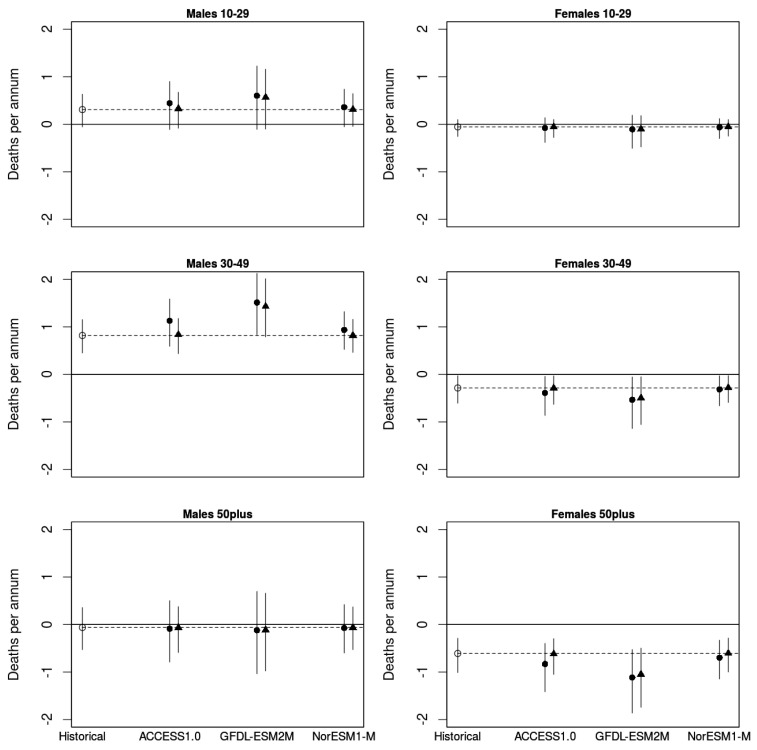
Numbers of suicide deaths per annum (and 95% CI) attributable to full drought conditions for each future GCM scenario (2000–2099) compared to historical rates (1971–1999); the right-hand triangular dots in each pair are RCP8.5 (high emissions scenario), while the left circular dots are RCP4.5 (lower emissions scenario).

**Table 1 ijerph-19-07855-t001:** Descriptive statistics of weather exposure data in rural NSW statistical divisions (SDs) during the historical period 1971–1999.

SD Group	Rain Annual Average (mm)	Monthly Maximum Temperature Annual Average (C)	Number of Full Droughts	Average Drought Duration (Months)	Maximum Drought Duration (Months)
Central West	605	22	7	11	25
Mid-North Coast	1309	22	8	10	14
Murray	410	23	6	10	16
Murrumbidgee	516	23	8	9	16
North and Far Western	371	26	6	9	12
Northern	760	23	4	11	19
Richmond-Tweed	1405	24	6	12	18
South Eastern	790	18	8	11	24

**Table 2 ijerph-19-07855-t002:** Descriptive statistics of suicides and population numbers in NSW SDs during the historical period 1971–1999.

SD Group	Sex	Total Suicides	Mean Annual Suicides	Mean Annual Population	Mean Annual Suicide Rate (per 100,000)
Central West	Male	434	15	67,747	22
	Female	105	4	67,248	6
Hunter	Male	1212	42	203,210	21
	Female	299	10	205,637	5
Illawarra	Male	809	28	132,224	21
	Female	228	8	131,181	6
Mid-North Coast	Male	530	18	83,073	22
	Female	114	4	85,065	5
Murray	Male	284	10	42,276	24
	Female	64	2	41,213	5
Murrumbidgee	Male	394	14	58,718	24
	Female	81	3	57,488	5
North and Far	Male	492	17	57,572	30
Western *	Female	74	3	55,804	5
Northern	Male	506	17	72,557	23
	Female	106	4	72,349	6
Richmond-	Male	434	15	62,366	24
Tweed	Female	109	4	64,221	6
South Eastern	Male	465	16	64,079	25
	Female	92	3	62,123	5
Sydney	Male	8758	302	1,424,283	21
	Female	3352	116	1,473,113	8

* North and Far Western SDs were combined to enable statistical modelling.

## Data Availability

Climate data are available on request from the authors. Suicide data from the Australian Causes of Death Unit Record File are available on request from the Australian Bureau of Statistics and are subject to approval by data custodians in the government.

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
