# Peer review of "Climate Change, Drought and Rural Suicide in New South Wales, Australia: Future Impact Scenario Projections to 2099"

_ijerph, 2022, doi:10.3390/ijerph19137855_

Round 1

Reviewer 1 Report

This is an interesting study with shows climat changes as predictable future. Changes (forest fire, flood, wind, drught, pollutions) are pointed out as predictable incidents in long perspective. You decided to present correlational research.

I am sure, you are proficient in climat changes, in statystic modeling. So the part Materials and methods apeal to me.

The problem is in explanation an facts. Accordance to Durkheim- there are facts, so what about argue.

Introduction

Q1. Do you have any theoretical reason to take off „Climat changes- drought” as specific (remarkable) factors for suicide?

There are a lot of dramatic situations incorrelate with suicide rate. For instance- In my country mountain wind in spring and autumn, All Saints’ Day in autumn, when we celerate the memory of dead relatives. But there is specific situation.

Ofcourse You pointed out pollutions problem („impacts of climate-change related air pollution”). Yes, but pollution is the factor- not trigger in suicide situation. It means lead in the air eliminated magnesium in our body. Consequently, day afther day, our natural skills to cope with stress reduce (there are suspicions like that).  It is a part of long process to siucide attack

On the other hand- COVID, economical transformation, war, there are in corelation with suicide rate. Not only climat- or drought.

Generally speaking- are You able to explain your model of research in theories?

Q2. Can You use any theories to explain your poin of view?

Much as I admire your invention. But as far me thet apaper needs more knowlege about process of suicide (not as a suicide rate).

What is the first: climat chages focus on mental and health condition? Or climat change may by only a trigger to suicide, providet that a man or women has individual determinats?

(„Droughts have been associated with distress and depression”, „drought on mental health”- could You explain your theretical poin of wiev?

Look, suicide in theories and research is describing as a process. It has phases, time, changes in thinking about myself, past and future. My quality research pointed out negative discribing ourself, siucide imagination, looking for the individual story of persons, who on previous dead in effect of suicide). It takes quite long time, and rarely attack is sudden. There are individual changes in our body, mind, social relations and so on. Ofcourse all of this may be disguisted.

   What is the role of climat changes: Is it a trigger, and it comes on blanded with may dfferent factors? Or - do you skip different factors, and dramatic evet (drought) is the reason of suicide?

Please, use some  theretical basis. Look, we are witnessing of climat events, but in general people cope with it.

Conclusions

Q3. Could You replay of your goal- „…can inform mitigation and adaptation strategies that will help prepare communities for the effects of climate change”

For IJERHP social aspect of research is very importat. For me too. I am educationalist, so as far I am concer, recomendation should be particular. It means-  who may protect people? Educationalists, psychologist, doctor, animators in social network? Local government in the regions of the country?

What should they do? Stop climat changes- I don’t know if it is possible. Protect inhabitants by prognostic in building policy? By eduaction, by promoting moving between part of country?

I don’t know. I approve of your opinion, the point is in specific local conditions.

Buy, as eduacationalist I expect to get to know something more. IJERHP promotes not only correlations of data, but prognostic role of science.

Please thing about your conclsion.  Now You skim the surface of practical, useful effect of research.    

Q4. Do you predict some differences between population (sex, age, role- rural women), but in subjective / individual context?

Is a chance to describe their specific life situation? Ofocurse accordance articles about this region? It os necessary to understand your results, the idea of prognostic recommendation.

Author Response

We thank the reviewer for their comments and below we have included a letter to explain, point by point, the details of the revisions to the manuscript and our responses to the referees’ comments.

Question 1. This is an interesting study with shows climate changes as predictable future. Changes (forest fire, flood, wind, drought, pollutions) are pointed out as predictable incidents in long perspective. You decided to present correlational research. I am sure, you are proficient in climate changes, and statistical modelling. So the part Materials and methods appeal to me. The problem is in explanation and facts. According to Durkheim- there are facts, so what about argue. 

Introduction: Do you have any theoretical reason to take off “Climate changes- drought” as specific (remarkable) factors for suicide?

There are a lot of dramatic situations that correlate with suicide rates. For instance- In my country mountain winds in spring and autumn, All Saints’ Day in autumn, when we celebrate the memory of dead relatives. But these are specific situations. Of course, You pointed out pollutions problem (“impacts of climate-change related air pollution”). Yes, but pollution is the factor- not trigger in suicide situation. It means lead in the air eliminated magnesium in our body. Consequently, day after day, our natural skills to cope with stress reduce (there are suspicions like that).  It is a part of long process to suicide attacks. On the other hand- COVID, economical transformation, war, these are in correlation with suicide rate. Not only climate- or drought. Generally speaking- are you able to explain your model of research in theories?

Answer 1.

We thank the reviewer for referring to the sociologist Durkheim. We agree that the implications of our observations are best served by sociological arguments and explanations. To take full advantage of your kind suggestion, we have added such sociological explanations with references into the discussion on lines 361-371. (e.g. on line 361 of the discussion we have now cited the paper [29] by Bryant LG, Garnham B. (2015). The fallen hero: masculinity, shame and farmer suicide in Australia. Gender Place Cult 22:67-82. Doi: 10.1080/09663 69X.2013.855628 and on line 363 we now cite the application of Durkheim’s work in the analysis in the paper [30] by Hogan A, Scarr E, Lockie S, Chantt B and Alston S. (2012). Ruptured Identity of Male Farmers: Subjective Crisis and the Risk of Suicide: Journal of Rural Social Sciences, 27(3), 118–140). 

Thank you for mentioning cultural and dramatic circumstances that are known to increase suicide rates in your country. Numerous studies support the relationship between suicide and hardship in other countries also. Here in Australia, droughts and extreme weather events are historically recognised for their considerable impacts on human health and wellbeing and we had originally cited some of these papers in our introduction (lines 34-35: “Droughts have been associated with distress and depression [8,9,10,11] and with suicide rates [12,13]”). Hence, to answer the first part of your question, we investigated drought and suicide because a key culturally ingrained observation of Australian climates is that of droughts and floods. Moreover, the science of climate change produces increasing certainty that these extremes will be exacerbated in the coming decades. Farming communities are the most vulnerable because they rely on moderate rainfall for their agricultural livelihoods. Our study was therefore designed around this direct link between climate and human productivity, with suicide deaths providing an extreme but hard metric for the inevitable mental health effects of droughts in agricultural communities.

In response to this and the reviewers second question we include a new paragraph in the introduction that presents the theoretical basis for this field of study (lines 50-68):

“There are several theories about how climatic drought may influence the suicide rate, as explored in previous review articles ([15] Dixon and Kalkstein 2009; [17] Vins et al. 2015). First, droughts increase financial stress on farmers and farming communities, even if partially compensated by drought relief welfare payments. Farmers are also more vulnerable to economic realities, such as rising interest rates, falling commodity prices and unfavourable foreign exchange rates. Reduced rainfall can directly depress economic activity in rural towns. In some regions the entire economy may be affected. Rural downturns can accelerate migration to metropolitan areas; weakening and stressing social support systems and inhibiting social interaction. In some cases rural depopulation may pass a tipping point, leading to an ongoing loss of critical services, such as hospitals, schools and doctors. Second, there can be a great psychological toll following environmental degradation and this may be acute during droughts, especially when decisions are made to sell or kill starving animals or to destroy orchards and vineyards, which in some cases were painstakingly accumulated over generations. Such loss, and even the apprehension of loss, undoubtedly places a burden on the mental health of farmers and their families. This mourning may not be confined to farmers because other sections of the community are also likely to be impoverished by long-term environmental degradation. The experience of seeing suffering wild plants and animals, or parched urban parks and gardens, and contemplation of their loss are likely to be extremely painful for some individuals.”

[15] Dixon P, Kalkstein A (2009) Climate-suicide relationships: A research problem in need of geographic methods and cross-disciplinary perspectives. Geography Compass 3:1–14. doi: 10.1111/j.1749-8198.2009.00286.x

[17] Vins, H., Bell, J., Saha, S. & Hess, J. (2015). The mental health outcomes of drought: A systematic review and causal process diagram. International Journal of Environmental Research and Public Health, 12(10), 13251–13275.

Our additional discussion (lines 358-371) is as follows:

“There are several possible explanations for the contrasting effects of drought on suicide rates among working males and females of the same Australian population. First, rural women seek diverse social supports that relieve stress whereas men may be too ashamed [29]. In an application of the suicide theories of Durkheim to farming communities, Hogan et al. suggests a male reluctance to adjust anomic and egoistic misconceptions in the face of identity shock [30]. Australian rural women may also be more stoic and resilient in the face of drought-related hardships. Furthermore, community support may strengthen as drought conditions deteriorate, reinforcing the support networks that benefit rural women most. In addition, the Australian government has previously offered financial support following drought declarations which may be beneficial for rural women but not for their male counterparts due to machismo [30]. Suicide rates are higher in farming communities than in urban areas in Australia [30-32] and in other countries [33]. In a review of this phenomena [31], it is argued that social, geographical and psychological factors together contribute to this disparity between rural and urban communities.”

Question 2. Can You use any theories to explain your point of view? Much as I admire your invention. But as for me the paper needs more knowledge about process of suicide (not as a suicide rate).  What is the first: climate changes focus on mental and health condition? Or climate change may by only a trigger to suicide, provided that a man or women has individual determinants?

(“Droughts have been associated with distress and depression”, “drought on mental health”- could You explain your theoretical point of view?)

Suicide in theories and research is described as a process. It has phases, time, changes in thinking about myself, past and future. My quality research pointed out negative describing ourself, suicide imagination, looking for the individual story of persons, who died of suicide). It takes quite long time, and rarely attack is sudden. There are individual changes in our body, mind, social relations and so on. Of course all of this may be discussed. What is the role of climate changes: Is it a trigger, and it comes on blended with many different factors? Or - do you skip different factors, and dramatic event (drought) is the reason of suicide? Please, use some theoretical basis. Look, we are witnessing of climate events, but in general people cope with it.

Answer 2. In the first question and this one, the reviewer distinguishes suicide-disposing factors from immediate triggers. While we appreciate the importance of this, our study took a whole population epidemiology approach that elucidates only the former. Under this framework of historical suicides linked with weather records and a sophisticated drought indicator (the Hutchinson Index), these correlational analyses identify significant effects of drought on suicide rates. Nonetheless, in full agreement with the reviewer, we have inserted the following paragraph to the discussion (lines 368-371) to provide additional theoretical basis and evidence from the literature that support our point of view:

“Suicide rates are higher in farming communities than in urban areas in Australia ([30] Hogan A, 2012, Kennedy A [31] 2014 and [32] 2020) and in other countries ([33] Stark C 2006)). In a review of this phenomena ([31] Kennedy A 2014), it is argued that social, geographical and psychological factors together contribute to this disparity between rural and urban communities.”

[30] Hogan A, Scarr E, Lockie S, Chantt B and Alston S. (2012). Ruptured Identity of Male Farmers: Subjective Crisis and the Risk of Suicide: Journal of Rural Social Sciences, 27(3), 118–140.

[31] Kennedy AJ, Maple MJ, McKay K, Brumby SA. (2014). Suicide and accidental death in Australia's rural farming communities: a review of the literature. Rural Remote Health.;14(1):2517. PMID: 24909931.

[32] Kennedy A, Adams J, Dwyer J, Rahman MA, Brumby S. (2020). Suicide in Rural Australia: Are Farming-Related Suicides Different? Int J Environ Res Public Health. 17(6):2010. doi: 10.3390/ijerph17062010.

[33] Stark C, Gibbs D, Hopkins P, Belbin A, Hay A, Selvaraj S. (2006). Suicide in farmers in Scotland. Rural Remote Health. 6(1):509. PMID: 16563050.

Question 3. Conclusions. Could You replay your goal- “…can inform mitigation and adaptation strategies that will help prepare communities for the effects of climate change”. For IJERPH social aspect of research is very important. For me too. I am educationalist, so as far I am concerned, recommendation should be particular. It means-  who may protect people? Educationalists, psychologist, doctor, animators in social network? Local government in the regions of the country? What should they do? Stop climate changes- I don’t know if it is possible. Protect inhabitants by prognostic in building policy? By education, by promoting moving between part of country? I don’t know. I approve of your opinion, the point is in specific local conditions. But, as educationalist I expect to get to know something more. IJERPH promotes not only correlations of data, but prognostic role of science. Please think about your conclusion.  Now you skim the surface of practical, useful effect of research.    

Answer 3. It is true that the observations of increased male and reduced female suicide rates under drought conditions in Australia fall short of prognosis. We now argue that these observations reflect a complex interplay of social structures and gender differences, possibly including normative gender roles. Our speculation about the rupture of the male identity in the face of drought induced crop failure or stock deaths could be addressed by correlating suicide rates with rural occupation data. These data would help to decipher the gender generalisations we make in the discussion (on lines 360-368). However such new data collection is outside the scope of our study. We have added the following additional text to the conclusions (lines 385-392) to present more specific recommendations for addressing this problem:

“This study alludes to a suite of contributing factors that influence suicide, drawn from the environmental, social and political context of life in Australia, of which drought is a part. These results help isolate the most critical times of risk, and could be used to better time and direct the deployment of social resources. This includes provision of targeted counselling services to vulnerable people, both during droughts and at times with hotter than average maximum temperatures. Other policy implications from this finding support broadening investment in research into gender specific drought effects rather than purely climate and economic focused research into drought impacts.”

Question 4. Do you predict some differences between population (sex, age, role- rural women), but in subjective / individual context? Is a chance to describe their specific life situation? Of course accordance articles about this region? It is necessary to understand your results, the idea of prognostic recommendation.

Answer 4.  Yes, we do predict differences between rural men and women but because we used population level data collection methods it is not possible to study the specific life situations of the suicide victims. Our study is therefore able to identify the burden of disease attributable to the broad-scale drought exposure data across the whole population and to use that information to assess the potential future scenarios. We used the available data to investigate this problem at the population level. We do agree, however, that additional information could be used to develop more individual prognostic recommendations to prevent suicides of farmers during droughts and further research projects should address this.

Reviewer 2 Report

A very solid and significant piece of work.  Well designed, good methodology and significant results,  Great work considering this issue from several viewpoints and in particular carefully considered possible confounding factors.  The supplementary information provides valuable data and text adding to the original article and providing a strong basis for the analysis provide in the paper.  very well done.

Author Response

We thank the reviewer for this positive comment.

Reviewer 3 Report

Before any further considerations, the authors should endeavour to explain their main results (increase in men and not women, rural and not urban population) based on available data and not on assumption. I believe that financial aspect may be a substantial bias for instance.

Author Response

Thank you for this comment. We agree that the implications of our observations should be supported by explanations. We have added more explanations with references into the discussion. For the present population level study, the available data were included for suicide rates in the entire population of New South Wales, Australia, and the latest climate change projections and scenarios, from which future droughts could be predicted. Our main observation is that suicide rates increase among men and decrease among women during droughts in Australia. Because this observation is true only in the rural context, it is likely that rural occupations, lifestyles and/or conditions are explanatory factors. Moreover, the gender difference further points to psychosocial determinants of mental health in these communities. We agree that in the absence of specialised and purpose-collected data such as a interviews with male and female suicide survivors from rural occupations, our results invite speculation about sex differences, gender roles and other psychosocial factors.    

Although we cannot provide such data within the scope of this study, we have improved the discipline of our interpretations by citing a range of co-varying factors, such as social support levels, normative masculinity roles and financial dependence on agriculture. For example on lines 332-338 we now more clearly refer to the systematic review of the literature by Vins et al. [15] who found that financial stresses on farming families can substantially undermine mental health. We have also inserted new discussion (lines 358-371) and citations that support the observed difference in suicide rates among rural male farmers from NSW (e.g on line 361 of the discussion we have cited the paper [29] by Bryant LG, Garnham B. (2015). The fallen hero: masculinity, shame and farmer suicide in Australia. Gender Place Cult 22:67-82. Doi: 10.1080/09663 69X.2013.855628 and on line 363 we cite the paper [30] by Hogan A, Scarr E, Lockie S, Chantt B and Alston S. (2012). Ruptured Identity of Male Farmers: Subjective Crisis and the Risk of Suicide: Journal of Rural Social Sciences, 27(3), 118–140.  Therefore, we have exhausted available data and can only encourage future studies and new data collections. Accumulation of such evidence will enable targeted mental health interventions that are tailored for the Australian rural context.

Round 2

Reviewer 3 Report

Line 296-302. 95 CI is not stated, please explain

How clinically significant are these results? The authors should use some kind of data representation to really emphasize how big is the difference (e.g NNT, although, I understand that use of risk models might limit the use of these parametrs). In line with this the authors should add as a supplement comparisons between suicide rates and different regions with thier respective indices of draught.

Author Response

Thankyou for noting our typographical errors that led to the omission of the 95% confidence intervals. We have corrected these on lines 296-302.

We also appreciate the thought that our results could be presented in terms of practical clinical significance. This led us to consider medical or even social interventions that, if targeted appropriately and in line with our findings, could reduce the suicide burden in agricultural communities. Suicide has to be distinguished, however, from conventionally treatable diseases because it does not always correlate with precedent mental health states. Some suicides are known to be a reactive response to a stressor without the presence of any underlying mental disorder (Ferrari AJ, Norman RE, Freedman G, et al. The burden attributable to mental and substance use disorders as risk factors for suicide: findings from the Global Burden of Disease Study 2010. PLoS One 2014; 9(4): e91936). The NNT refers to treatment efficacy. In this case the only plausible treatment would be the pre-emptive relocation of all middle aged male farmers into urban settings and away from their farms, which is an untenable proposal. 

To emphasize a supplementary comparison between suicide rates and droughts in different regions in line with the reviewer’s suggestion, we now present the practical significance of the results by stating that: “The predicted number of annual suicides in middle aged males attributable to periods of full drought accounted for around 2% of the total number of suicides in that group in rural regions, compared with 0% in urban regions.” (Line 303-306). All suicides present a substantial social burden and should be avoided, therefore the drought impact is of practical significance.